# Efficacy and Safety of Chemotherapy Combined with Hormonal Therapy in Heavily Pretreated Advanced Epithelial Ovarian, Fallopian Tube, or Primary Peritoneal Cancer (ELSA/KGOG3049): A Multicenter Pilot Study

**DOI:** 10.3390/cancers17142320

**Published:** 2025-07-12

**Authors:** Kidong Kim, Chel Hun Choi, Sang-Yoon Park, Min Kyu Kim, Keun Ho Lee, Eun-Ju Lee, Myong Cheol Lim, Young Han Park, Min Sun Kyung, Jae Hong No, Dong Hoon Suh, Jeong-Won Lee, Sangjeong Ahn, Banghyun Lee

**Affiliations:** 1Department of Obstetrics and Gynecology, Seoul National University Bundang Hospital, 82 Gumi-ro, 173 Beon-gil, Bundang-gu, Seongnam-si 13620, Gyeonggi-do, Republic of Korea; kidong.kim.md@gmail.com (K.K.); nojaehong@naver.com (J.H.N.); sdhwcj@naver.com (D.H.S.); 2Department of Obstetrics and Gynecology, Samsung Medical Center, Sungkyunkwan University College of Medicine, 81 Irwon- ro, Gangnam-gu, Seoul 06351, Republic of Korea; chelhun.choi@samsung.com (C.H.C.); garden.lee@samsung.com (J.-W.L.); 3Center for Gynecologic Cancer, National Cancer Center, 323 Ilsan-ro, Ilsandong-gu, Goyang-si 10408, Gyeonggi-do, Republic of Korea; parksang@ncc.re.kr (S.-Y.P.); mclim@ncc.re.kr (M.C.L.); 4Department of Obstetrics and Gynecology, Samsung Changwon Hospital, Sungkyunkwan University of Medicine, 158, Paryong-ro, Masanhoewon-gu, Changwon-si 51353, Gyeongsangnam-do, Republic of Korea; seoulminkyukim@gmail.com; 5Department of Obstetrics and Gynecology, Seoul St. Mary’s Hospital, The Catholic University of Korea, 222 Banpo-daero, Seochogu, Seoul 06591, Republic of Korea; hohoho@catholic.ac.kr; 6Department of Obstetrics and Gynecology, Chung-Ang University School of Medicine, Chung-Ang University Hospital, 102 Heukseokro, Dongjak-gu, Seoul 06973, Republic of Korea; ejlee@cau.ac.kr; 7Department of Obstetrics and Gynecology, Hallym University Sacred Heart Hospital, University of Hallym College of Medicine, 22 Gwanpyeong-ro, 170 beon-gil, Dongan-gu, Anyang-si 14068, Gyeonggi-do, Republic of Korea; drparkyh@yahoo.co.kr; 8Department of Obstetrics & Gynecology, Hallym University Dongtan Sacred Heart Hospital, University of Hallym College of Medicine, 7 Keunjaebong-gil, Hwasung-si 18450, Gyeonggi-do, Republic of Korea; msfeel@hallym.or.kr; 9Department of Pathology, Korea University Anam Hospital, Korea University College of Medicine, 73 Goryeodae-ro, Seongbuk-gu, Seoul 02841, Republic of Korea; vanitasahn@gmail.com; 10Department of Obstetrics and Gynecology, Inha University Hospital, Inha University College of Medicine, 27 Inhang-ro, Sinheung-dong, Jung-gu, Incheon 22332, Republic of Korea

**Keywords:** estrogen receptor, high-grade serous ovarian cancer, megestrol acetate, ovarian cancer, progesterone receptor, tamoxifen

## Abstract

Many patients with advanced epithelial ovarian, fallopian tube, or primary peritoneal cancer experience recurrence after the initial treatment, and the effectiveness of chemotherapy decreases with each recurrence. Therefore, new strategies are urgently needed to improve outcomes in patients who have already undergone multiple lines of treatment. Hormonal therapy has been used for ovarian cancer, but its role when combined with chemotherapy and tailored to hormone receptor expression is unclear. This study examined whether combining chemotherapy with hormonal therapy could improve clinical outcomes in patients with heavily pretreated cancer. Patients whose tumors had more estrogen receptors received tamoxifen, while those with more progesterone receptors received megestrol acetate, each combined with chemotherapy. These findings suggest that the combination of tamoxifen and chemotherapy may provide potential clinical benefit and manageable safety in selected patients. These exploratory results support a further investigation of a personalized approach based on hormone receptor status in late-line ovarian cancer therapy.

## 1. Introduction

Approximately 75% of patients with epithelial ovarian, fallopian tube, or primary peritoneal cancer (EOC) present with advanced-stage disease at diagnosis [1]. The standard therapy for EOC is staging surgery, including complete debulking surgery and platinum-based chemotherapy. Although the response rate to initial treatment is high, up to 85% of patients who achieve complete remission experience recurrence following first-line chemotherapy [2]. The efficacy of second- or later-line chemotherapy remains unsatisfactory [2,3]. Resistance to chemotherapy increases whenever EOC recurs [2,3]. The lack of methods to overcome chemotherapy resistance is the main reason for the poor prognosis in EOC.

Many studies have reported using hormonal receptors (HR) as prognostic biomarkers for EOC, despite results being inconsistent because of small sample sizes, discrepancies in antibodies used for immunohistochemistry, and inconsistent analysis methods [4,5,6,7]. Previous studies reported that the estrogen receptor (ER) and progesterone receptor (PR^a^) were expressed in 32–100% and 19–91% of EOC cases, respectively [8,9]. In addition, ER-α and ER-β are encoded by different genes, which have proliferative and anti-proliferative effects [9,10]. Previous studies reported that PR^a^ mediates the growth inhibitory effect of progesterone and induces apoptosis [9,10].

Some studies with and without HR status evaluations have shown that a combination of tamoxifen or medroxyprogesterone acetate (MPA) and platinum-based chemotherapy in advanced or recurrent EOC is safe and may improve the chemotherapy response through a synergistic interaction [11,12]. In contrast, randomized controlled trials (RCTs) using tamoxifen or MPA combined with platinum-based chemotherapy as the first-line therapy for advanced EOC found that combination therapies were safe but failed to improve survival compared to platinum-based chemotherapy alone [13,14]. Nevertheless, in these studies, tamoxifen or MPA was not used based on HR expression [13,14]. Moreover, phase II, single-arm, prospective studies have shown that LHRH agonists linked to doxorubicin and androgen receptor (AR) inhibitors improve survival in recurrent EOC that expresses the LHRH receptor and AR, respectively [15,16].

EOC encompasses various types of cancers based on the histology and genetic findings [1]. Therefore, tailored therapy based on the cancer type is needed instead of the current uniform treatment approach. HR expression differs according to histological type in EOC [4]. A previous study reported that ER and PR^a^ are expressed in 60.7% and 31.2% of high-grade serous ovarian cancers (HGSCs), respectively [4].

No studies have analyzed the efficacy and safety of chemotherapy combined with hormonal therapy according to HR expression in various histologic types of EOC. In heavily pretreated advanced EOC, this study hypothesized that chemotherapy combined with the tailored targeting of hormonal therapies according to HR expression could be a novel strategy to improve the chemotherapy response through synergistic interactions. Therefore, this study examined the efficacy and safety of physician-chosen chemotherapy with hormonal therapy according to HR expression in patients with heavily pretreated advanced EOC.

## 2. Materials and Methods

### 2.1. Study Design and Participants

ELSA, the Korean Gynecologic Oncology Group (KGOG) 3049, was a phase II, multicenter pilot study conducted at nine centers in South Korea. All participating centers were secondary or tertiary hospitals that regularly performed surgical care for ovarian cancer and had multidisciplinary teams, including specialized gynecologic oncologists, radiologists, and pathologists.

The eligible criteria were as follows: age 19 years or older, EOC histologically confirmed from cytoreductive surgery (upfront or interval), (two previous chemotherapy regimens and just previous progression-free interval < six months) or (≥ three previous chemotherapy regimens), Eastern Cooperative Oncology Group performance status of 0–2, measurable lesion on imaging (CT or MRI) or increase in CA125, an estimated life expectancy of at least six months, and adequate hematologic and end-organ function. The following patients were excluded: those diagnosed with or treated for other types of primary cancer within the past five years and patients undergoing anticancer therapy that included immunotherapy or other targeted therapies. Initially, the study protocol required patients to undergo surgery or biopsy immediately after the last recurrence to assess the current HR status. Nevertheless, the eligibility criteria were amended to permit enrollment based on previously available surgical or biopsy tissue because of the extremely low enrollment rate over the first year.

The study protocol was approved by the ethics committees at all participating institutions. All patients provided written informed consent. This study was conducted in accordance with the Bioethics and Safety Act of the Republic of Korea and was registered with the Clinical Research Information Service (CRIS; registration number KCT0004571) on 20 December 2019.

### 2.2. Procedures

HR expression in ovarian cancer tissue was assessed by conducting immunohistochemistry in the central laboratory using ‘primary surgery tissue’ or ‘tissue obtained from surgery or biopsy after recurrence’ if primary surgery tissues are unavailable (Figure 1). In patients with the primary surgery tissue and tissues obtained after recurrence, the last acquired tissue was used to assess HR expression. Two pathologists performed the central review using the Allred scoring system to assess HR expression. Allred scoring [17] uses the sum of intensity score (on a scale of 0–3) and proportion score (on a scale of 0–5). A total score of 0–2 was considered negative HR expression, while a score of 3–8 was considered positive.

The HR status in each patient was categorized as ER-dominant, PR^a^-dominant, or no receptor expression, based on the relatively higher Allred score after comparing ER and PR^a^ expression (Figure 1). In cases where the ER and PR^a^ Allred scores were equal and positive, the patients were alternatively allocated to one of the dominant groups. Patients with no positive expression for either receptor were excluded from the study.

The ER-dominant patients (who had ER expression alone or a relatively higher ER expression rate) received oral tamoxifen (20 mg twice daily) and physician-chosen chemotherapy (Figure 1). The PR^a^-dominant patients (who had PR^a^ expression alone or a relatively higher PR^a^ expression rate) received oral megestrol acetate (MA) (160 mg once daily) and physician-chosen chemotherapy (Figure 1). The physician-chosen chemotherapy included belotecan, docetaxel, etoposide, gemcitabine, paclitaxel, pegylated liposomal doxorubicin, topotecan, and vinorelbine. Radiologists in the participating centers assessed tumors on CT or MRI, and individual investigators assessed the response of combined therapies according to “Revised RECIST Guideline (Version 1.1) [18].” The investigators evaluated the adverse events according to the Common Terminology Criteria for Adverse Events (CTCAE) v5.0 [19]. The association between adverse events and treatment drugs was assessed as unknown, unrelated, possible, probable, or definite.

The patients continuously received hormonal therapies from day 1 of the first cycle of each physician-chosen chemotherapy until progressive disease (PD) or the development of unacceptable toxicity, or for at least six months. Based on the response assessment performed at six months, patients showing a complete response (CR), partial response (PR^b^), or stable disease (SD) received continuous hormonal therapies until PD or the development of unacceptable toxicity, regardless of the continuation of physician-chosen chemotherapy if the patients wanted to take them. After a six-month response assessment, physician-chosen chemotherapy may be performed according to the investigators’ discretion.

The study was conducted independently in two arms: the ER-dominant group and the PR^a^-dominant group (Figure 1).

The dose modifications and treatment discontinuation for tamoxifen, MA, or physician-chosen chemotherapy were permitted to manage adverse events. The doses were reduced by 25%, and up to two reductions were allowed. The patient had a dose reduction twice, but was excluded from the study if the dose had to be reduced again. Within three weeks after discontinuing treatment, the patients could restart the treatment drugs at the investigators’ discretion, but these patients were excluded from the study if the treatment drugs could not be restarted.

The patients were followed up every eight weeks (±two weeks) during the combined therapies for six months. After a six-month response assessment, the patients were followed up every three months until PD, development of unacceptable toxicity, or the end of this study. This study ended three months after the final enrolled patient received combined therapies for six months. CT or MRI imaging and laboratory evaluations were performed at each follow-up visit, and adverse events were monitored. These evaluations, including imaging studies, laboratory data, and adverse event profiles, were assessed according to the RECIST and CTCAE criteria. All patients were followed up to finally assess the adverse events related to hormonal therapy at three months after discontinuing tamoxifen or MA.

### 2.3. Outcomes

In the ER-dominant arm, the effects of the combined therapy of tamoxifen and physician-chosen chemotherapy were evaluated. In the PR^a^-dominant arm, the effects of combined therapy of MA and physician-chosen chemotherapy were evaluated. The outcomes of the two arms were not compared (Figure 1).

The best objective response rate (ORR) for six months was assessed as the primary outcome (Figure 1). The best response among the three response assessments performed at two-month intervals was selected. CR or PR^b^ was confirmed when those were repeated at the response assessment after two months. The secondary outcomes were as follows: best ORR according to the histologic types of EOC, time to progression (TTP), and adverse events. TTP was defined as the duration from the initiation of combination therapy to the date of radiologic or clinical disease progression.

### 2.4. Sample Size

The optimal two-stage Simon design was used (Figure 2).

The response probabilities for the poor and good drugs were 0.05 and 0.20, respectively. The one-sided Type I error rate and power were 5% and 80%, respectively.

This study was conducted in accordance with the following principles. In the first stage, 10 patients were enrolled. The study was terminated if no patient demonstrated a response. The study proceeded to the second stage if more than one patient responded. In the second stage, an additional 19 patients were enrolled, bringing the total to 29 patients. The study was terminated if fewer than four patients showed a response. If four or more patients responded, the study was considered successful, and the null hypothesis was rejected. Hence, further investigation was warranted.

The best objective response for six months was used to assess the response. The development of unacceptable toxicity from tamoxifen or MA was considered a non-response. The study was performed in the ER-dominant and PR^a^-dominant arms. The planned number of patients was 58 (29 patients in the ER-dominant arm and 29 patients in the PR^a^-dominant arm).

The dropout rate was not initially accounted for in the study design, which planned to enroll patients continuously until the predefined number in each arm (29 patients) had either experienced PD, developed unacceptable toxicity, or completed the six-month response assessment. A high dropout rate was observed as the study neared completion, prompting the study team to enroll additional patients even before the planned number of patients in each arm had reached a study endpoint. This decision was made to prevent prolonged delays in study completion. The main reasons for dropout included withdrawal of consent, deterioration of the subjects’ general condition, allergic reactions to chemotherapy, and refusal to take oral medication due to ileus or other treatment-limiting conditions. As a result, four more patients than initially planned were enrolled in the ER-dominant arm and included in the final analysis. The final proportion of patients who reached a study endpoint (PD, unacceptable toxicity, or completion of the six-month response assessment) was 66.7% (36 out of 54).

Although the planned number of patients was 29 per arm, accrual in the PR^a^-dominant arm remained extremely low throughout the study. Because enrollment in the ER-dominant arm proceeded as planned, the data and safety monitoring committee (DSMC) permitted the study to continue without protocol amendment. As a result, only three patients were ultimately enrolled in the PR^a^-dominant arm.

### 2.5. Statistics

Descriptive statistics were used to summarize the treatment outcomes. The categorical variables were reported as frequencies and percentages. The best ORR, ORR, clinical benefit rate (CBR), and PD were calculated with exact binomial 95% confidence intervals (CIs). The Kaplan–Meier method was used to estimate the probability of progression over time. The results are presented as TTP probability curves. The 95% CIs for the Kaplan–Meier estimates were calculated using the log–log transformation method. Patients without progression at the time of analysis were censored at the date of the last follow-up. All statistical analyses were conducted using R version 4.4.1 (R Foundation for Statistical Computing, Vienna, Austria), using the ′survival′ and ′survminer′ packages.

## 3. Results

The study was conducted between 1 September 2019 and 15 August 2024. In the ER-dominant arm, three patients showed responses among the first 10 enrolled patients in the first stage, and nine patients showed responses among the total 33 patients during the second stage. During the second stage, four additional patients were enrolled before the planned number of 19 patients had either experienced PD, developed unacceptable toxicity, or completed the six-month response assessment—the planned number required for the evaluation. One patient was allocated to the ER-dominant arm because the Allred scores for ER and PR^a^ were equal. Consequently, 33 patients were included in the ER-dominant arm, and the efficacy and safety outcomes were analyzed. In the PR^a^-dominant arm, only three patients were enrolled during the entire study period of the ER-dominant arm. Therefore, this study in the PR^a^-dominant arm ended without additional patient enrollments. Four patients who had mucinous carcinoma (one patient), clear cell carcinoma (CCC) (two patients), and HGSC (one patient) dropped out of the study because no ER and PR^a^ expression was detected.

### 3.1. Baseline Characteristics

Table 1 lists the baseline characteristics of the 36 study subjects. Thirty-three patients showed ER-dominant expression, and three showed PR^a^-dominant expression.

All patients with ER-dominant expression had high-grade serous ovarian cancer (HGSC) and received combination therapy with tamoxifen at the third to ninth lines of chemotherapy. The lines of chemotherapy were as follows: seven patients in the third line, eight patients in the fourth line, five patients in the fifth line, five patients in the sixth line, two patients in the seventh line, four patients in the eighth line, and two patients in the ninth line. Sixteen different chemotherapeutic agents were used among the 33 patients with ER-dominant expression. Each agent was used in one to five patients, and the distribution of agents was as follows: cyclophosphamide (five patients); pegylated liposomal doxorubicin, topotecan, and vinorelbine (four patients each); belotecan, topotecan/carboplatin, topotecan/cisplatin, and weekly paclitaxel (two patients each); and eight other agents were used in single patients. Moreover, the tissue samples for testing HR expression were obtained from primary surgery (29 patients), surgery at the first recurrence (one patient), biopsy at the second recurrence (one patient), and surgery at the fifth recurrence (two patients).

Patients with PRa-dominant expression had CCC (two patients) and HGSC (one patient). They received combination therapy with MA at the third line (one patient), fourth line (one patient), and seventh line (one patient) of chemotherapy. Three different chemotherapy agents were used across three patients: docetaxel/carboplatin, topotecan, and vinorelbine. The tissue for testing HR expression was obtained from primary surgery (two patients) and surgery at the second recurrence (one patient).

### 3.2. Response Assessment in Patients with ER-Dominant Expression

The best ORR for six months was (9/33) 27.3%. It consisted of (1/33) 3.0% of CR and (8/33) 24.2% of PR^b^ (Table 2).

The patient with CR received combination therapy with tamoxifen at the fourth line of chemotherapy (case 23). The patient showed CR in the two- and four-month response assessments and dropped out after a four-month response assessment. Therefore, the patient response was included in the best ORR for six months but not in the ORR at six months (Table 1, Table 2 and Table 3). The rates of PR^b^ decreased according to the increase in the chemotherapeutic lines (Table 3).

At six months, the CR, PR^b^, and SD were 0% (0/32), 18.8% (6/32), and 18.8% (6/32), respectively. Therefore, at six months, the ORR and CBR were 18.8% (6/32) and 37.5% (12/32), respectively. PD occurred in 62.5% (20/32) within six months. The TTP was distributed as follows: one month (two patients), two months (four patients), four months (five patients), five months (one patient), and six months (eight patients) (Table 2 and Figure 3). According to the increase in chemotherapeutic lines, the PR^b^ and SD rates decreased, and the PD rates increased (Table 3).

After a six-month response assessment, nine patients received tamoxifen maintenance therapy with or without chemotherapy until PD (seven patients) or dropout (two patients) occurred. The maximum time until PD occurred was 14 months. Moreover, the maximum times when the patients were assessed as CR, PR^b^, and SD were nine, 10, and 13 months, respectively. In three patients with PR^b^ at the six-month response assessment, the CR or PR^b^ responses during tamoxifen maintenance therapy were as follows: CR at nine months and PD at 12 months after combined therapy at the third line of chemotherapy (case 20), PR^b^ at 10 months and PD at 14 months after combined therapy at the fourth line of chemotherapy (case 2), and PR^b^ at eight months and PD at 11 months after combined therapy at the third line of chemotherapy (case 28) (Table 1 and Table 2, and Figure 3).

Among the nine patients who achieved the best ORR and the six patients who achieved the ORR at six months, nine and six different chemotherapeutic agents, respectively, were used (Table 1 and Table 2)

### 3.3. Response Assessment in Patients with PR^a^-Dominant Expression

The best ORR for six months was 0% (0/3). At six months, ORR and CBR were 0% (0/3), and PD occurred in 100% (3/3) of cases until six months (Table 4). Two patients with CCC who received combination therapy with MA at the fourth or seventh line of chemotherapy showed PD at two months (cases 34 and 35). A patient with HGSC who received combination therapy with MA at the third line of chemotherapy showed PD at six months (case 36) (Table 1 and Table 4).

### 3.4. Adverse Events

No unacceptable toxicity related to tamoxifen or MA was shown. Of the 36 patients with ER-dominant expression, only one patient showed a possible adverse event (grade two nausea) within one month after she received a combination therapy of tamoxifen with paclitaxel/cisplatin at the fourth line of chemotherapy (case 31). No adverse events occurred in three patients with PR^a^-dominant expression (Table 1 and Appendix A).

## 4. Discussion

This study was performed on patients with heavily pretreated advanced EOC. The ER-dominant patients received tamoxifen and physician-chosen chemotherapy, and the PR^a^-dominant patients received MA and physician-chosen chemotherapy. The best ORR for six months was assessed as the primary outcome. All 33 patients with ER-dominant expression had only HGSC. A favorable best ORR was observed in ER-dominant patients, with no unacceptable toxicity related to tamoxifen. By contrast, three patients with PR^a^-dominant expression had CCC and HGSC histology and progressed until six months. Unacceptable toxicity related to MA was not shown.

Several small studies examined the effects of the combination therapy of chemotherapy and tamoxifen or MPA in EOC. In these studies, however, hormonal therapy was not performed according to HR expression [11,12,13,14]. In a retrospective study without an evaluation of the HR status (n = 50), the platinum-based chemotherapy and tamoxifen combination induced an ORR of 50% in relapsed or progressive advanced EOC after platinum-based chemotherapy failed [11]. In an RCT, however, where primary adjuvant chemotherapy for advanced EOC was performed, and ER and PR^a^ were evaluated in 72% of patients, a combination of platinum-based chemotherapy and tamoxifen (n = 49) showed similar survival rates to those of platinum-based chemotherapy (n = 51) without a correlation between therapy and HR [13]. A retrospective study of primary adjuvant chemotherapy for advanced EOC showed that a combination of platinum-based chemotherapy and MPA (n = 22) induced higher 10-year survival rates than platinum-based chemotherapy alone (n = 28), suggesting better survival with higher PR^a^ expression [12]. By contrast, in another RCT (n = 71) where primary adjuvant chemotherapy for advanced EOC was performed, and HR was not evaluated, the overall remission and survival rates were similar in the groups given platinum-based chemotherapy alone or platinum-based chemotherapy in combination with either MPA or 5-fluorouracil [14]. In this study, where tailored hormonal therapy was conducted according to ER- or PR^a^-dominant expression, the combination therapy of physician-chosen chemotherapy and tamoxifen showed encouraging responses in patients with ER-dominant expression. Nevertheless, the findings are limited because of the lack of a comparator and the small sample size. Conversely, it was difficult to obtain valuable information from the combination therapy of physician-chosen chemotherapy and MA because of the extremely small number of patients.

The ORR of chemotherapy in patients with recurrent EOC ranges from 3% to 53% [2,20]. Moreover, advanced lines of chemotherapy beyond second-line chemotherapy in EOC are associated with low response [20]. A retrospective study reported the following ORRs in patients with recurrent EOC (n = 156): 51.6%, 11.9%, 2.9%, 4.5%, and 0% for second-, third-, fourth-, fifth-, and ≥ sixth-line chemotherapy, respectively [20]. A drastic decline in the clinical response rates of chemotherapy was observed in those who received advanced lines of chemotherapy. Hence, various therapeutic strategies are needed to overcome these in patients with heavily pretreated advanced EOC. Targeted therapies may be useful therapeutic strategies in heavily pretreated advanced EOC. Recent studies reported that the ORR of targeted therapy ranged from 32.4% to 63.6% in patients with advanced or recurrent EOC with high folate receptor α or human epidermal growth factor 2 expression [21,22,23]. This study evaluated the effects of chemotherapy combined with targeted hormonal therapies according to ER or PR^a^ expression in patients with heavily pretreated advanced EOC. The combination of chemotherapy and tamoxifen, administered as third- to ninth-line therapy, improved the clinical response of patients with ER-dominant expression, suggesting a synergistic effect of the combined treatment. According to the increase in the chemotherapeutic lines of patients with ER-dominant expression, the number of patients enrolled decreased, and the responses decreased, showing low responses in advanced lines of chemotherapy beyond the fourth-line chemotherapy. Moreover, in this study, a few patients who received combined therapy at the third or fourth line of chemotherapy and received tamoxifen maintenance therapy after PR^b^ at the six-month response assessment showed a good response and delay of disease progression, suggesting a prolonged synergistic effect of combined therapy and a beneficial effect of tamoxifen maintenance therapy in relatively early chemotherapeutic lines.

HGSC, the most common ovarian cancer subtype, accounts for 63.4% of EOC [24,25]. A large-scale ovarian tumor tissue analysis consortium study revealed the following HR expression: HGSC (n = 1610) (81% of ER, 31% of PR^a^, and 16% of no ER and PR^a^); CCC (n = 354) (19.2% of ER, 7.9% of PR^a^, and 79% of no ER and PR^a^); and mucinous carcinoma (n = 185) (20.5% of ER, 15.7% of PR^a^, and 77% of no ER and PR^a^) [4]. In this study, the participants were enrolled prospectively according to the eligibility criteria regardless of the histologic subtype of EOC. Nevertheless, all patients with ER-dominant expression had HGSC (n = 33), and patients with PR^a^-dominant expression had CCC (n = 2) and HGSC histology (n = 1). Moreover, patients without ER and PR^a^ expression who dropped out had mucinous carcinoma (n = 1), CCC (n = 2), and HGSC (n = 1). Corresponding to previous studies [4,24,25], the results showed that HGSC was the most common histologic subtype of EOC and was associated with relatively high ER expression. In contrast, CCC and mucinous carcinoma did not show ER and PR^a^ expression at high frequency. Moreover, CCC was associated with relatively high PR^a^ expression. These findings revealed the beneficial effects of the chemotherapy combined with tamoxifen in 33 HGSC patients with ER-dominant expression and the poor effect (PD at two months) of chemotherapy combined with MA in two CCC patients with PR^a^-dominant expression.

Many studies have reported that tamoxifen and progestin therapy (MA and MPA) are associated with a low incidence of general and severe toxicity in gynecologic cancers [13,26,27,28,29,30]. Corresponding to previous studies [13,26,27,28,29,30], in the present study, no toxicity attributable to tamoxifen or MA was observed in patients with ER- or PR^a^-dominant expression, except for one case of mild toxicity.

Previous studies comparing ER and PR^a^ expression in primary and recurrent HGSC patients reported controversial discordance rates [31,32]. One study (n = 107) reported non-significant discordance rates of 34% and 12.4% in ER and PR^a^, respectively, with higher ER and lower PR^a^ expression in recurrent cancers compared to primary cancers [31]. Another study (n = 80) showed a non-significant discordance rate of 8.2% in ER (lower ER expression in recurrent cancers) and a significant discordance rate of 15.7% in PR^a^ (lower PR^a^ expression in recurrent platinum-sensitive cancers) [32]. HR expression assessments using the tissue of recurrent cancers are suitable for targeted hormonal therapy because HR expression may differ between primary and recurrent cancers. Nevertheless, in this study, only three patients underwent surgery or biopsy immediately before enrollment, and most did not receive surgery or biopsy after recurrence. Therefore, most HR expression assessments were performed using the tissues from primary surgery.

In this study, although physician-chosen chemotherapy regimens were used in combination with tamoxifen or MA, the risk of selection bias was limited. In the ER-dominant arm, 16 different chemotherapeutic agents were administered across 33 patients. Each agent was used in only one to five patients, and among the nine patients with the best ORR and six patients with ORR at six months, no specific agent was disproportionately represented. Similarly, each of the three patients in the PR^a^-dominant arm received a different chemotherapeutic regimen. This distribution suggests that there was no clustering of responses to specific agents. Moreover, as this was a multicenter pilot study involving heavily pretreated patients, treatment flexibility was essential to reflect real-world clinical practice and facilitate adequate accrual. Therefore, although treatment heterogeneity exists, its impact on the treatment efficacy appears minimal.

This study was the first to investigate the efficacy and safety of chemotherapy combined with hormonal therapy based on HR expression and the histologic types in EOC. This study had the following limitations. First, in the PR^a^-dominant arm, only three patients were enrolled throughout the study period, precluding a valuable assessment of the treatment efficacy. Although all three patients showed PD after combination therapy with MA, it remains unclear whether this reflects the true resistance or is simply due to insufficient data. Future studies with larger PR^a^-dominant cohorts are needed. Second, HR expression was primarily assessed using tissue from primary surgery because most patients did not undergo surgery or a biopsy after recurrence. This may not accurately reflect the receptor status at the time of treatment. Third, the ER-dominant arm lacked a control group and had a relatively small sample size, limiting the generalizability of the findings. The use of physician-chosen chemotherapy regimens also introduced treatment heterogeneity. Nevertheless, this heterogeneity had a limited impact on the outcomes, considering the broad distribution of agents and absence of clustering among responders. Therefore, these findings should be interpreted with caution and validated in larger, controlled studies with standardized treatment protocols.

## 5. Conclusions

This multicenter pilot study examined the efficacy and safety of physician-chosen chemotherapy combined with tamoxifen or MA based on ER- or PR^a^-dominant expression in patients with heavily pretreated advanced EOC. In patients with ER-dominant expression, chemotherapy combined with tamoxifen showed encouraging clinical activity and favorable safety in heavily pretreated advanced HGSC patients. Nevertheless, these findings require validation in larger, controlled trials given the exploratory nature, small sample size, and lack of a control group in this study. In contrast, the extremely small number of patients with PR^a^-dominant expression precluded valuable assessments of the treatment efficacy. Future large-scale studies are warranted to determine the clinical utility of combining chemotherapy with hormonal therapy based on HR expression in this setting.

## Figures and Tables

**Figure 1 cancers-17-02320-f001:**
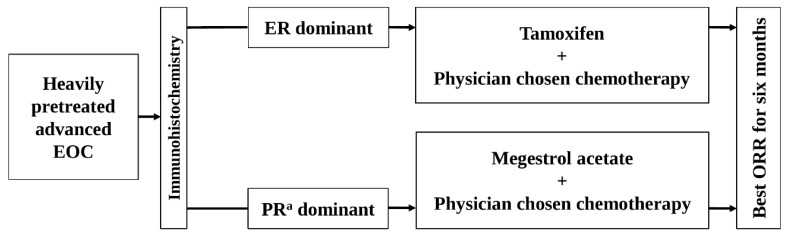
The schema of this study. EOC, epithelial ovarian cancer; ER, estrogen receptor; ORR, objective response rate; PR^a^, progesterone receptor.

**Figure 2 cancers-17-02320-f002:**
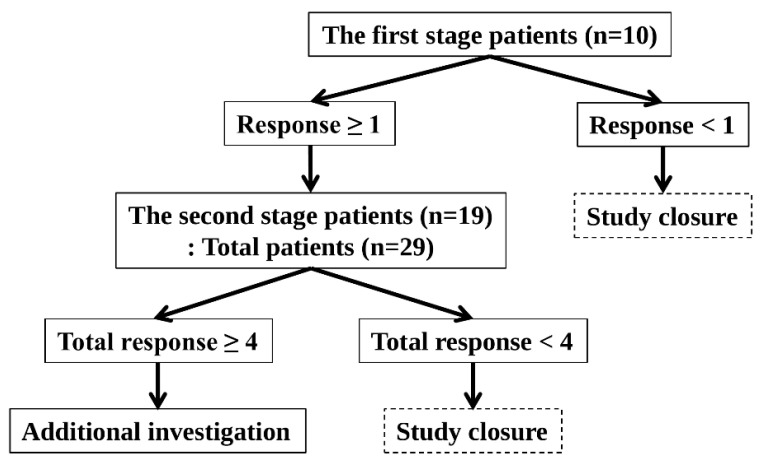
Optimal Two-Stage Simon Design.

**Figure 3 cancers-17-02320-f003:**
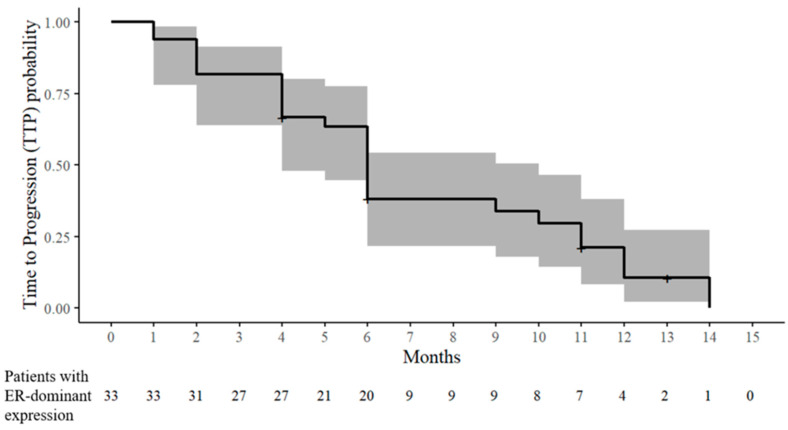
Kaplan–Meier estimate of the time to progression (TTP) probability in patients with ER-dominant expression. The solid line denotes the estimated probability of remaining progression-free, and the shaded band represents the 95% CIs. The tick marks indicate censored observations. The numbers at risk are displayed below the *x*-axis.

**Table 1 cancers-17-02320-t001:** Baseline characteristics.

	Age	Stage	Histology	Previous Chemotherapeutic Line	Current Chemotherapeutic Line	Method for Obtaining Tissue for Testing HR Expression	Tissue Used for Testing HR Expression	Status of HR Expression (Allred Scores)	Status of HR Expression	Hormonal Agent	Chemotherapeutic Agent
Case 1	65	3C	HGSC	2nd	3rd	Primary surgery	Ovary	ER-dominant (ER: 7, PR^a^: 5)	ER-dominant	Tamoxifen	Belotecan
Case 2	63	4B	HGSC	3rd	4th	Primary surgery	Ovary	ER-dominant (ER: 8, PR^a^: 3)	ER-dominant	Tamoxifen	Weekly paclitaxel
Case 3	55	3A	HGSC	7th	8th	Primary surgery	Ovary	ER-dominant (ER: 7, PR^a^: 0)	ER-dominant	Tamoxifen	Cyclophosphamide
Case 4	66	3C	HGSC	7th	8th	Primary surgery	Ovary	ER-dominant (ER: 4, PR^a^: 0)	ER-dominant	Tamoxifen	Vinorelbine
Case 5	44	3B	HGSC	8th	9th	Primary surgery	Ovary	ER-dominant (ER: 7, PR^a^: 2)	ER-dominant	Tamoxifen	Cyclophosphamide
Case 6	62	3C	HGSC	7th	8th	Primary surgery	Fallopian tube	ER-dominant (ER: 7, PR^a^: 3)	ER-dominant	Tamoxifen	Cyclophosphamide
Case 7	70	3C	HGSC	5th	6th	Primary surgery	Peritoneum	ER-dominant (ER: 5, PR^a^: 4)	ER-dominant	Tamoxifen	Vinorelbine
Case 8	59	3A	HGSC	8th	9th	Surgery at 1st recurrence	Pelvic lymph node	ER-dominant (ER: 7, PR^a^: 5)	ER-dominant	Tamoxifen	Cyclophosphamide
Case 9	57	4B	HGSC	5th	6th	Primary surgery	Ovary	ER-dominant (ER: 8, PR^a^: 4)	ER-dominant	Tamoxifen	Cyclophosphamide
Case 10	48	3C	HGSC	6th	7th	Primary surgery	Omentum	ER-dominant (ER: 7, PR^a^: 6)	ER-dominant	Tamoxifen	Docetaxel/carboplatin
Case 11	74	3C	HGSC	7th	8th	Primary surgery	Ovary	ER-dominant (ER: 7, PR^a^: 5)	ER-dominant	Tamoxifen	Vinorelbine
Case 12	53	3C	HGSC	2nd	3rd	Primary surgery	Ovary	ER-dominant (ER: 5, PR^a^: 5)	ER-dominant	Tamoxifen	Weekly gemcitabine
Case 13	66	4B	HGSC	6th	7th	Primary surgery	Ovary	ER-dominant (ER: 7, PR^a^: 0)	ER-dominant	Tamoxifen	Vinorelbine
Case 14	50	4B	HGSC	4th	5th	Primary surgery	Ovary	ER-dominant (ER: 7, PR^a^: 6)	ER-dominant	Tamoxifen	Topotecan/carboplatin
Case 15	61	2A	HGSC	3rd	4th	Primary surgery	Ovary	ER-dominant (ER: 6, PR^a^: 3)	ER-dominant	Tamoxifen	Gemcitabine
Case 16	65	4B	HGSC	3rd	4th	Primary surgery	Ovary	ER-dominant (ER: 7, PR^a^: 3)	ER-dominant	Tamoxifen	Weekly paclitaxel
Case 17	71	3B	HGSC	3rd	4th	Primary surgery	Mesentery	ER-dominant (ER: 7, PR^a^: 5)	ER-dominant	Tamoxifen	Topotecan
Case 18	50	4A	HGSC	3rd	4th	Primary surgery	Colon	ER-dominant (ER: 7, PR^a^: 6)	ER-dominant	Tamoxifen	Topotecan/carboplatin
Case 19	51	3C	HGSC	5th	6th	Primary surgery	Ovary	ER-dominant (ER: 7, PR^a^: 4)	ER-dominant	Tamoxifen	Docetaxel triweekly
Case 20	63	3C	HGSC	2nd	3rd	Primary surgery	Ovary	ER-dominant (ER: 8, PR^a^: 4)	ER-dominant	Tamoxifen	Topotecan/cisplatin
Case 21	55	4B	HGSC	2nd	3rd	Primary surgery	Ovary	ER-dominant (ER: 7, PR^a^: 4)	ER-dominant	Tamoxifen	Pegylated liposomal doxorubicin
Case 22	41	3B	HGSC	3rd	4th	Primary surgery	Ovary	ER-dominant (ER: 6, PR^a^: 4)	ER-dominant	Tamoxifen	Weekly topotecan
Case 23	49	3C	HGSC	3rd	4th	Primary surgery	Ovary	ER-dominant (ER: 4, PR^a^: 3)	ER-dominant	Tamoxifen	Topotecan
Case 24	67	4B	HGSC	2nd	3rd	Primary surgery	Salpinx	ER-dominant (ER: 6, PR^a^: 0)	ER-dominant	Tamoxifen	Pegylated liposomal doxorubicin
Case 25	50	3C	HGSC	2nd	3rd	Primary surgery	Ovary	ER-dominant (ER: 5, PR^a^: 4)	ER-dominant	Tamoxifen	Topotecan/cisplatin
Case 26	51	3C	HGSC	5th	6th	Surgery at the 5th recurrence	Spleen	ER-dominant (ER: 8, PR^a^: 7)	ER-dominant	Tamoxifen	Pegylated liposomal doxorubicin
Case 27	60	3C	HGSC	5th	6th	Surgery at the 5th recurrence	Paraaortic lymph node	ER-dominant (ER: 8, PR^a^: 2)	ER-dominant	Tamoxifen	Gemcitabine/carboplatin
Case 28	65	3C	HGSC	2nd	3rd	Biopsy at the 2nd recurrence	Supraclavicular lymph node	ER-dominant (ER: 7, PR^a^: 4)	ER-dominant	Tamoxifen	Belotecan
Case 29	56	3C	HGSC	4th	5th	Primary surgery	Rectum	ER-dominant (ER: 7, PR^a^: 3)	ER-dominant	Tamoxifen	Pegylated liposomal doxorubicin
Case 30	63	4B	HGSC	4th	5th	Primary surgery	Ovary	ER-dominant (ER: 7, PR^a^: 5)	ER-dominant	Tamoxifen	Pegylated liposomal doxorubicin/carboplatin
Case 31	59	3C	HGSC	3rd	4th	Primary surgery	Ovary	ER-dominant (ER: 6, PR^a^: 4)	ER-dominant	Tamoxifen	Paclitaxel/cisplatin
Case 32	70	3C	HGSC	4th	5th	Primary surgery	Ovary	ER-dominant (ER: 7, PR^a^: 6)	ER-dominant	Tamoxifen	Topotecan
Case 33	55	1C	HGSC	4th	5th	Primary surgery	Ovary	ER-dominant (ER: 6, PR^a^: 5)	ER-dominant	Tamoxifen	Topotecan
Case 34	65	3B	Clear cell	3rd	4th	Surgery at the 2nd recurrence	Liver	PR^a^-dominant (ER: 0, PR^a^: 6)	PR^a^-dominant	Megestrol acetate	Topotecan
Case 35	49	4B	Clear cell	6th	7th	Primary surgery	Ovary	PR^a^-dominant (ER: 2, PR^a^: 3)	PR^a^-dominant	Megestrol acetate	Vinorelbine
Case 36	60	3C	HGSC	2nd	3rd	Primary surgery	Ovary	PR^a^-dominant (ER: 4, PR^a^: 5)	PR^a^-dominant	Megestrol acetate	Docetaxcel/carboplatin

ER, estrogen receptor; HGSC, high-grade serous ovarian cancer; HR, hormonal receptor; PR^a^, progesterone receptor.

**Table 2 cancers-17-02320-t002:** Response assessment in patients with ER-dominant expression.

	Response for 6 Months				Response After 6 Months
	CR	PR^b^	SD	PD	
Case 1			2, 4, and 6 months		SD at 9 months and PD at 11 months ^a^
Case 2		**2, 4, and 6 months**			PR at 10 months and PD at 14 months ^a^
Case 3			2 months	4 months	
Case 4			2 and 4 months	6 months	
Case 5			2 and 4 months	6 months	
Case 6			2 months	4 months	
Case 7				2 months	
Case 8			2 months	4 months	
Case 9			2 and 4 months	6 months	
Case 10			2, 4, and 6 months		Dropout after SD at 9 and 11 months ^a^
Case 11				2 months	
Case 12		2 months	4 months	6 months	
Case 13			2 months	4 months	
Case 14				2 months	
Case 15			2 months	4 months	
Case 16			2, 4, and 6 months		Drop out ^c^
Case 17		2 months	4 and 6 months		SD at 9 months and PD at 12 months ^a^
Case 18		**2 and 4 months**		6 months	
Case 19				2 months	
Case 20		**2, 4, and 6 months**			CR at 9 months and PD at 12 months ^a^
Case 21		**4 and 6 months**	2 months		Drop out ^c^
Case 22				1 month	
Case 23 ^d^	**2 and 4 months**				
Case 24			2 months	5 months	
Case 25			2 and 4 months	6 months	
Case 26			2, 4, and 6 months		PD at 10 months ^a^
Case 27		**4 and 6 months**	2 months		Drop out ^c^
Case 28 ^e^		**2 and 6 months**	4 months		PR at 8 months and PD at 11 months ^a^
Case 29			2 and 4 months	6 months	
Case 30		**2 and 6 months**			Dropout after SD at 10 and 13 months ^a^
Case 31	**4 months**	**2 months**	6 months		PD at 9 months ^a^
Case 32			4 months	6 months	
Case 33				1 month	
**Best ORR for 6 months**		**(9/33) 27.3% (95% CI, 12.1–42.5)**			
ORR at 6 months		(6/32) 18.8% (95% CI, 5.2–32.3)			
CBR at 6 months		(12/32) 37.5% (95% CI, 20.7–54.3)			
PD until 6 months		(20/32) 62.5% (95% CI, 45.7–79.3)			

CBR, clinical benefit rate; CI, confidence interval; CR, complete response; ER, estrogen receptor; ORR, objective response rate; PD, progressive disease; PR^b^, partial response; SD, stable disease. The bold words represent the best objective response. The underlined words represent SD at six months. ^a^ Tamoxifen was administered until PD or dropout occurred. ^c^ Drop out after a six-month response assessment. ^d^ In case 23, the patient dropped out after a four-month response assessment. Therefore, the response of the patient was included in the best ORR for six months, whereas it was not included in the ORR at six months. ^e^ In case 28, PR^b^ was documented at both the two- and six-month assessments, whereas SD was observed at the four-month evaluation. In accordance with the response assessment criteria requiring confirmation, the best ORR was determined to be PR^b^.

**Table 3 cancers-17-02320-t003:** Response assessment by line of chemotherapy in patients with ER-dominant expression.

	Chemotherapy Line						
	3rd	4th	5th	6th	7th	8th	9th
Total patients with the best objective response assessment for 6 months (n = 33)	7/33 (100%)	8/33 (100%)	5/33 (100%)	5/33 (100%)	2/33 (100%)	4/33 (100%)	2/33 (100%)
Patients with the best objective response for 6 months (n)	3/7 (42.9%)	4/8 (50%)	1/5 (20%)	1/5 (20%)	0 (0%)	0 (0%)	0 (0%)
CR	0 (0%)	1/8 (12.5%) ^a^	0 (0%)	0 (0%)	0 (0%)	0 (0%)	0 (0%)
PR^b^	3/7 (42.9%)	3/8 (37.5%)	1/5 (20%)	1/5 (20%)	0 (0%)	0 (0%)	0 (0%)
Total patients with response assessment at 6 months (n = 32)	7/32 (100%)	7/32 (100%)	5/32 (100%)	5/32 (100%)	2/32 (100%)	4/32 (100%)	2/32 (100%)
Patients with objective response at 6 months (n)	3/7 (42.9%)	1/7 (14.3%)	1/5 (20%)	1/5 (20%)	0 (0%)	0 (0%)	0 (0%)
CR	0 (0%)	0 (0%)	0 (0%)	0 (0%)	0 (0%)	0 (0%)	0 (0%)
PR^b^	3/7 (42.9%)	1/7 (14.3%)	1/5 (20%)	1/5 (20%)	0 (0%)	0 (0%)	0 (0%)
Patients with SD at 6 months (n)	1/7 (14.3%)	3/7 (42.9%)	0 (0%)	1/5 (20%)	1/2 (50%)	0 (0%)	0 (0%)
Patients with PD until 6 months (n)	3/7 (42.9%)	3/7 (42.9%)	4/5 (80%)	3/5 (60%)	1/2 (50%)	4/4 (100%)	2/2 (100%)

CR, complete response; ER, estrogen receptor; PD, progressive disease; PR^b^, partial response; SD, stable disease. ^a^ In case 23, the patient showed CR in two- and four-month response assessments and dropped out after a four-month response assessment. Therefore, the patient was included in patients with the best objective response for six months, whereas she was not included in patients with the objective response at six months.

**Table 4 cancers-17-02320-t004:** Response assessment in patients with PR^a^-dominant expression.

	Response for 6 Months			
	CR	PR^b^	SD	PD
Case 34				2 months
Case 35				2 months
Case 36			2 and 4 months	6 months
Best ORR for 6 months			0%	
ORR at 6 months			0%	
CBR at 6 months			0%	
PD until 6 months			(3/3) 100%	

CBR, clinical benefit rate; CR, complete response; ORR, objective response rate; PD, progressive. disease; PR^a^, progesterone receptor; PR^b^, partial response; SD, stable disease.

## Data Availability

The original contributions presented in this study are included in the article/Appendix A. Further inquiries can be directed to the corresponding author(s).

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
