# Peer review of "Efficacy and Safety of Chemotherapy Combined with Hormonal Therapy in Heavily Pretreated Advanced Epithelial Ovarian, Fallopian Tube, or Primary Peritoneal Cancer (ELSA/KGOG3049): A Multicenter Pilot Study"

_cancers, 2025, doi:10.3390/cancers17142320_

Round 1
Reviewer 1 Report
Comments and Suggestions for Authors
The manuscript entitled Efficacy and Safety of Chemotherapy Combined with Hormonal Therapy in Heavily Pretreated Advanced Epithelial Ovarian, Fallopian Tube, or Primary Peritoneal Cancer (ELSA/KGOG3049): A Multicenter Pilot Study is a well-designed and timely that supports a personalized approach to advanced EOC management. The study addresses a pressing clinical challenge: the limited efficacy of chemotherapy in heavily pretreated advanced epithelial ovarian cancer (EOC). It explores a novel approach by tailoring hormonal therapy (tamoxifen or megestrol acetate) based on hormone receptor expression, potentially contributing to more personalized and effective treatments. The multicenter design increases external validity.
My comments are as follows: Most HR assessments were performed on primary tumor tissue rather than recurrent disease, which may not reflect current receptor status. The “physician-chosen chemotherapy” introduces heterogeneity, which could confound the interpretation of treatment effects. The justification for assigning patients to ER- or PR-dominant arms when both receptors were expressed is underexplained.
The lack of response in the PR-dominant group limits conclusions for that cohort. Data are not presented graphically (Kaplan-Meier plots), which would help visualize response and progression times.
My suggestion is to correct the inaccuracies and to have a language revision.
Comments on the Quality of English Language
The manuscript is generally understandable and conveys the intended scientific content clearly, the English could benefit from moderate editing to improve readability, precision, and professional polish.
Reviewer 2 Report
Comments and Suggestions for Authors
The authors present a phase II multicenter pilot study evaluating chemotherapy combined with hormone therapy (tamoxifen or megestrol acetate) in heavily pretreated patients with advanced pithelial Ovarian, Fallopian Tube, or Primary Peritoneal Cancer, stratified by hormone receptor status. I include the following comments below.
Although the study protocol planned to include 29 patients per arm, the PRa-dominant arm was closed after the inclusion of only three patients, without a solid methodological justification. Reasons such as low recruitment rate, lack of eligibility, protocol amendments, or decisions from a monitoring committee are not detailed, which represents a significant inconsistency between the planned design and the actual execution of the study.
The definition of the 'ER-dominant' and 'PRa-dominant' arms is not clearly operationalized. The quantitative criteria used to determine 'relative dominance' between ER and PRa when both were expressed are not specified. Furthermore, no objective threshold is established to define dominance. I suggest greater clarity in the stratification criteria.
The presentation of the number of included patients is confusing. Although the design called for 29 patients per arm, a total of 33 patients were ultimately analyzed in the ER-dominant arm without a detailed methodological explanation. Additionally, one patient with ambiguous ER/PRa expression was included, introducing a subjective decision in treatment allocation. A total of 36 patients are reported, but this progression from 29 to 33 and then to 36 is not clearly explained in the methodology, nor is it stated whether the analysis followed an intention-to-treat or per-protocol approach. I recommend clarifying this from the methodology section and reflecting it clearly in Figure 2.
The results are relevant, showing an ORR of 27.3% in the ER-dominant arm among patients receiving third- to ninth-line treatment, along with an absence of significant toxicity. However, clinical outcomes are reported only as raw proportions, without confidence intervals (CIs), which limits the assessment of their statistical precision.
In Table 3, it is recommended to improve the presentation of the n and percentage data. Currently, values are shown in parentheses, supposedly representing ‘n’, when in fact they correspond to percentages. This could lead to misinterpretation. It would be clearer to separate the two values (e.g., 3/7 (42.9%)) or to use two distinct columns for n and %.
Additionally, the title of Table 3 does not adequately reflect its contents, which not only shows the distribution by chemotherapy line, but also the associated clinical outcomes (ORR, CR, PR, SD, PD). I suggest revising the title to more accurately represent the analytical purpose of the table.
I also recommend conducting a Kaplan–Meier analysis to estimate time to progression or duration of response, especially since longitudinal data are available, including events of progression at different time points and tamoxifen maintenance treatments. The inclusion of survival curves would have allowed a more robust characterization of clinical benefit over time.
The PRa-dominant arm was closed with only three patients, making any conclusions about the ineffectiveness of MA premature.
In some analyses, n=33 is used, in others n=32, and in some cases, nine responses are reported but only six are counted for the six-month ORR. This inconsistency is somewhat confusing.
Regarding the discussion, I believe the interpretation of the results in the ER-dominant arm is overstated. It is not methodologically sound to claim a significant clinical benefit from tamoxifen plus chemotherapy based solely on a 27.3% ORR, especially in the absence of a control group, comparative analyses, or multivariable adjustments.
I suggest discussing the potential selection bias related to physician-chosen chemotherapy regimens, and whether this may have led to heterogeneity in treatment efficacy.
In the conclusion, I suggest approaching some ideas with caution, for example, "chemotherapy combined with tamoxifen demonstrated significant clinical benefit and favorable safety." But not before demonstrating the relevant statistical analyses.
Reviewer 3 Report
Comments and Suggestions for Authors
Our surgical colleagues conducted a study on epithelial ovarian carcinoma and primary peritoneal carcinoma. Their goal was to take stock of the current situation with a prospective multicenter study. The abstract summarizes the entire paper well and pushes the reader to understand the reasoning behind this work. The next section introduces the topic well and we really liked the sentence "The standard therapy for EOC is surgical staging, which includes complete cytoreduction and platinum-based chemotherapy". We also do not agree with this type of decision. It is true that patients who have this problem often come to the doctor's observation for complications of the disease or for significant ascites. At this time the oncological pathway must clarify the exact pathology with imaging diagnostics, ascitic sampling to study the cells, neoplastic markers and possible exploratory laparoscopy to calculate the PCI. In light of this, the most appropriate systemic therapeutic decisions must be taken in the multidisciplinary commission, as also described in the materials and methods, including, for both peritoneal and ovarian tumors, also PIPAC which allows the most appropriate chemotherapy to be aerosolized in the abdomen and which is now part of the systemic therapeutic treatments. However, we must not lose sight of the nutritional treatment of these patients who fail early and this could then influence the chemotherapy treatment. (doi.org/10.3390/nu17010188 to be read and cited in the bibliography). The results obtained by colleagues are good and introduce a discussion, also supported by a good bibliography. The sentence that best defines the study is: "In this study, where personalized hormonal therapy was conducted based on the dominant expression of ER or PRa, the combination therapy of physician-selected chemotherapy and tamoxifen showed significant responses, although it was difficult to obtain valuable information from the combination therapy of physician-selected chemotherapy and MA due to the extremely small number of patients". We also agree that the strategies in this type of pathology must be personalized to the patient. Moreover, generalizing, we are going, in the oncological therapy of each neoplasm, towards a personalization of the therapy. We agree with the authors' conclusions. Good iconography and excellent English
Round 2
Reviewer 2 Report
Comments and Suggestions for Authors
The authors made the reviewers' suggestions to the manuscript.